**Data Availability Statement:** All included studies can be searched and downloaded from their official websites or PubMed. 1. Doi:10.1016/s1470-2045

# A Bayesian network meta-analysis of the primary definitive therapies for locoregionally advanced nasopharyngeal carcinoma: IC +CCRT, CCRT+AC, and CCRT alone

Zhan-Jie Zhang[1], Liang-Liang Shi[1], Xiao-Hua Hong[1], Bo-Ya Xiao[2], Guo-He Lin[3], Quentin Liu[4], Bi-Cheng Wang[1] *

**1** Cancer Center, Union Hospital, Tongji Medical College, Huazhong University of Science and Technology, Wuhan, China, **2** Eastern Hepatobiliary Surgery Hospital, Second Military Medical University, Shanghai, China, **3** Department of Oncology, the Second Affiliated Hospital of Anhui Medical University, Hefei, China, **4** State Key Laboratory of Oncology in South China, Collaborative Innovation Center for Cancer Medicine, Cancer Center, Sun Yat-sen University, Guangzhou, China

* bcsnowell@163.com

## Abstract

### Background

The major aim of this Bayesian network analysis was to determine the optimal treatment strategy for locoregionally advanced nasopharyngeal carcinoma (LANPC).

### Method

We systematically searched databases and extracted data from randomized clinical trials involving LANPC patients randomly assigned to receive induction chemotherapy followed by concurrent chemoradiotherapy (IC+CCRT), CCRT followed by adjuvant chemotherapy (CCRT+AC), or CCRT.

### Results

In the network analysis, IC+CCRT was significantly better than CCRT alone for 5-year FFS (odds ratio [OR]: 1.63, 95% credible interval [CrI] 1.16–2.29), DMFS (OR: 1.56, 95% CrI 1.08–2.22), and LFRS (OR: 1.62, 95% CrI 1.02–2.59), but not OS (OR: 1.35, 95% CrI 0.92–2.00). Rank probabilities showed that IC+CCRT was ranked the best followed by CCRT+AC and CCRT for all 5-year outcomes. Although compared to IC+CCRT and CCRT, CCRT+AC did not significantly improve survival but had the highest 5-year survival rates.

### Conclusions

IC+CCRT could be recommended as a front-preferred primary definitive therapy for patients with LANPC.

(11)70320-5. 2. Doi:10.1016/j.ejca.2017.01.002. 3. Doi:10.1016/s1470-2045(16)30410-7. 4. Doi:10.1093/annonc/mdr116. 5. Doi:10.1016/j.ijrobp.2015.01.002. 6. Doi:10.1093/annonc/mdx770. 7. Doi:10.1093/annonc/mdy249. 8. Doi:10.1056/NEJMoa1905287. 9. Doi:10.1002/ijc.32099. 10. Doi:10.1016/j.ejca.2016.12.039. 11. Doi:10.1016/j.ejca.2019.07.007. 12. Doi:10.1002/cncr.29208.

**Funding:** This study was supported by the Hubei Provincial Natural Science Foundation (Grant number: 2020CFB397 to Bi-Cheng Wang) and the Independent Innovation Foundation of Wuhan Union Hospital (Grant number: 2019-109 to Bi-Cheng Wang). The funders had no role in study design, data collection and analysis, decision to publish, or preparation of the manuscript.

**Competing interests:** The authors have declared that no competing interests exist.

## Introduction

Previous phase III clinical trials have confirmed that concurrent chemoradiotherapy (CCRT) is superior to radiotherapy (RT) for patients with locoregionally advanced NPC (LANPC) [1–3]. Therefore, in the 2011 and 2012 National Comprehensive Cancer Network (NCCN) guidelines, CCRT was suggested as a category 1 recommendation [4,5].

Since 2013, the category of evidence supporting CCRT for LANPC has been cut to 2B [6]. At the same time, CCRT followed by adjuvant chemotherapy (CCRT+AC) was given a 2A recommendation [6]. However, the cited studies do not fully support the use of CCRT+AC over CCRT, as the two trials demonstrated the superiority of CCRT+AC compared to RT but not CCRT [1,7]. Additionally, another two prospective clinical trials failed to show that the addition of AC to CCRT improved survival outcomes [8–10].

From 2015, more prospective and large randomized studies have illustrated the benefit of induction chemotherapy followed by CCRT (IC+CCRT) versus CCRT alone in the treatment of LANPC [11]. However, until 2018, the level of evidence for IC+CCRT in the NCCN guideline was adjusted from category 3 to category 2A [12]. Our recently published study also demonstrated that, compared with CCRT, IC+CCRT significantly reduced the risks of death in patients with LANPC (3-year OS hazard ratio [HR]: 0.70, 95% confidence interval [CI] 0.55–0.89; 5-year OS HR: 0.77, 95% CI 0.62–0.94) [13]. To date, when LANPC patients receive IC, gemcitabine/cisplatin or docetaxel/cisplatin/5-fluorouracil could be recommended as category 1 regimens [14,15].

According to the 2020 NCCN guideline, IC+CCRT, CCRT+AC, and CCRT alone are all the primary definitive therapies for LANPC. Nevertheless, the cited studies recommending the application of CCRT+AC have not been updated. Since the absence of a randomized trial directly comparing IC+CCRT to CCRT+AC and to explore the optimal therapeutic strategy for LANPC, we conducted this Bayesian network analysis to comprehensively compare the efficacies of IC+CCRT, CCRT+AC, and CCRT.

## Methods

This study was conducted based on the Preferred Reporting Items for Systematic Reviews and Meta-analyses (PRISMA) extension statement for network meta-analysis [16].

### Search strategy

A systematic search of articles was conducted on PubMed, EMBASE, Web of Science, and Cochrane Library for clinical trials comparing at least any two of the three types of treatments on Nov 1, 2020. All the identified trials and relevant reviews were identified through reference lists to ensure completeness. The search terms in PubMed, EMBASE, and Web of Science online databases were "induction OR neoadjuvant OR adjuvant" and "concurrent OR concomitant" and "cisplatin" and "chemotherapy OR chemoradiotherapy OR radiotherapy" and "nasopharyngeal" and "carcinoma OR cancer OR tumor" and "study OR trial", while the search terms used in searching Cochrane Library included: "induction OR neoadjuvant OR adjuvant" and "concurrent OR concomitant" and "cisplatin" and "chemotherapy OR chemoradiotherapy OR radiotherapy" and "nasopharyngeal carcinoma OR nasopharyngeal cancer OR nasopharyngeal tumor".

### Selection criteria

Eligible trials were requested to satisfy the following "PICO" inclusion criteria: (P) patients were newly diagnosed with LANPC; (I+C) LANPC patients randomly received at least two of

the three treatment strategies, including IC+CCRT, CCRT+AC, and CCRT; (O) full-text articles and data of survival rates were available. Additional criteria comprised: (1) CCRT was cisplatin-based conventional concomitant chemoradiotherapy; (2) trials should be officially registered prospective phase II-IV clinical studies; (3) target therapy was prohibited during the whole care process; (4) published language was English. Any discrepancies were resolved by discussion.

## Outcomes

The primary endpoints were the rates of 5-year overall survival (OS) and failure-free survival (FFS). The secondary endpoints were 5-year rates of distant metastasis-free survival (DMFS) and locoregional recurrence-free survival (LRFS).

## Data extraction and risk of bias assessment

Bi-Cheng Wang and Zhan-Jie Zhang independently extracted the survival rates of all outcomes and the data of basic study characteristics and treatment modalities. All raw data were directly collected from the eligible clinical trials without further modification or adjustment. Review Manager 5.3 software (Cochrane Collaboration's Information Management System) was used to assess the risk of bias.

## Data synthesis and statistical analysis

Bayesian network meta-analyses were built using the Bayesian Markov Chain Monte-Carlo (MCMC) sampling in WinBUGS or GeMTC. The random-effects model was proposed. The effect size was estimated with odds ratios (ORs) and 95% credible intervals (CrIs) using the available raw data extracted from the trials. ORs were computed on averages of the 100,000 iterations (four chains with a thinning interval of 10) after a training phase of 50,000 burn-ins. Node-splitting approach was performed to evaluate if there was inconsistency in the closed-loop. Pairwise meta-analyses were applied to evaluate incoherence between direct and indirect evidence. The probability of each treatment being the best among the three therapies was estimated by using the distribution of the ranking probabilities.

Pooled OS, FFS, DMFS, and LRFS rates and 95% CIs were analyzed by using STATA 14.0 software. The heterogeneity between rates was tested by $I^2$ statistic percentages. Whether the heterogeneity was high ($I^2 > 50\%$) or low ($I^2 \leq 50\%$), random effects were performed to reduce the bias.

## Results

### Basic information on eligible clinical trials

Through a literature search, 1664 records were identified, of which 802 were duplicated records and 838 were excluded based on screening titles and abstracts (**Fig 1**). After a full-text article assessment of 121 remaining studies, 12 studies within 9 randomized clinical trials were included in this Bayesian network analysis [9–11,17–25].

The basic characteristics of all eligible trials were listed in **Table 1**. A total of 3140 randomly assigned LANPC patients were involved: 1321 received IC+CCRT, 411 received CCRT+AC, and 1408 received CCRT. 2/3-dimensional radiotherapy (2/3DRT) and intensity-modulated radiotherapy (IMRT) had been applied in these trials. The regimens of induction chemotherapy comprised cisplatin+epirubicin+paclitaxel, gemcitabine+carboplatin+paclitaxel, docetaxel+cisplatin+5-fluorouracil, mitomycin C+epirubicin+cisplatin+5-fluorouracil, gemcitabine+cisplatin, and cisplatin+5-fluorouracil. The adjuvant chemotherapy in both selected trials was

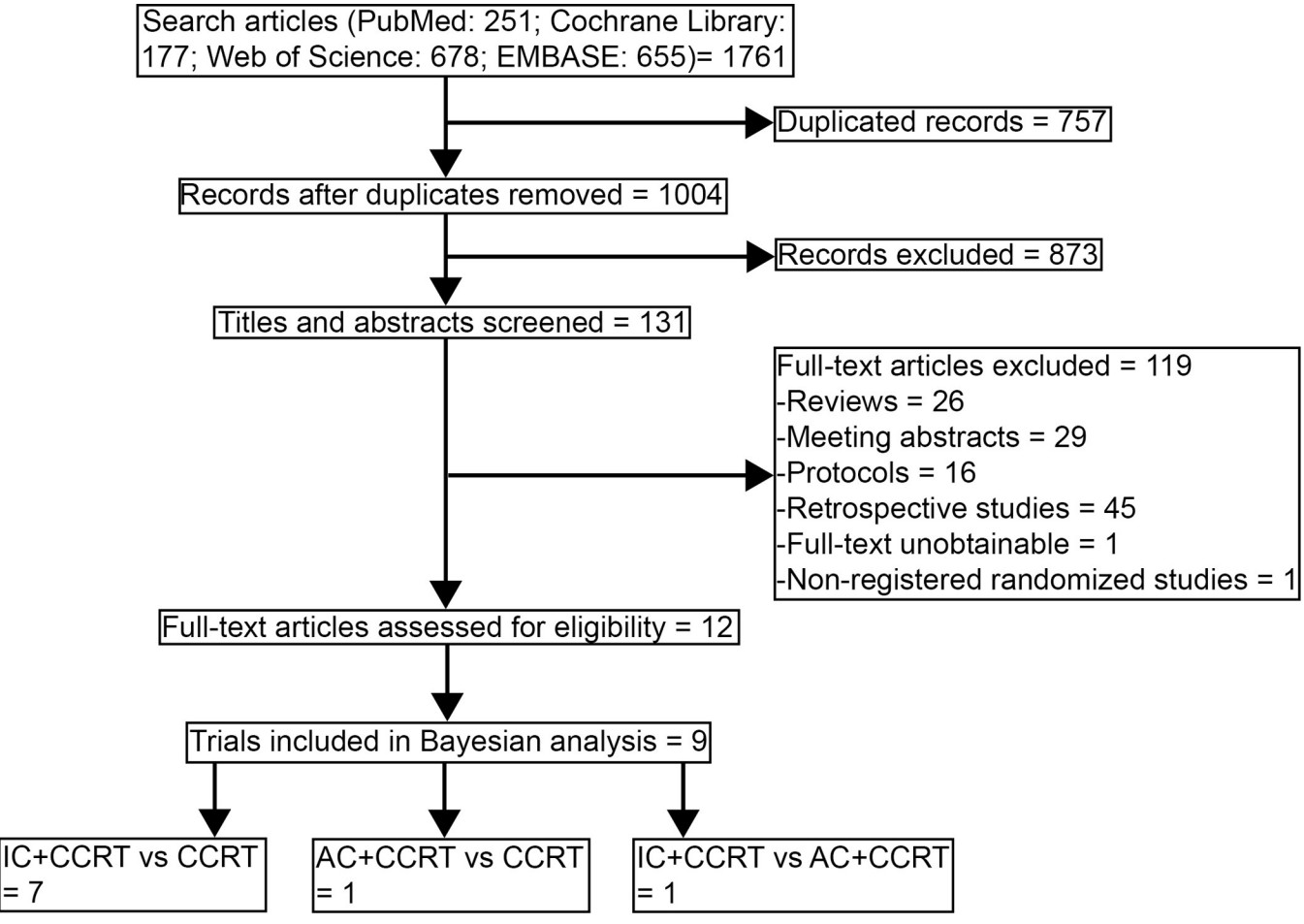

**Fig 1. Flowchart of literature search and selection.**

cisplatin+5-fluorouracil. In NPC-0501 trial, we extracted data of two arms (patients received cisplatin plus 5-fluorouracil IC or AC) in order to reduce the risk of bias [25]. In addition, the concurrent chemotherapies included weekly and triweekly cisplatin strategies.

Although all eligible trials were well-randomized studies, the methodological quality of this analysis was at moderate risk for reporting bias owing to the open-label design (**S2 Fig**).

### Bayesian network analyses of efficacy

We established a network to compare IC+CCRT, CCRT+AC, and CCRT (**S1 Fig**). **Fig 2** summarizes the Bayesian analyses results for 5-year rates data of OS, FFS, DMFS, and LRFS. According to the results, IC+CCRT was statistically superior to CCRT in terms of 5-year FFS (OR 1.63, 95% CrI: 1.16–2.29), DMFS (OR 1.56, 95% CrI: 1.08–2.22), and LRFS (OR 1.62, 95% CrI: 1.02–2.59) but not statistically superior for OS (OR 1.35, 95% CrI: 0.92–2.00). CCRT+AC showed no significant difference in assessed outcomes compared with IC+CCRT and CCRT alone, and there were no significant differences for all survival outcomes.

Ranking on 5-year survival indicated that IC+CCRT had the highest probabilities (OS: 62%; FFS: 82%; DMFS: 62%; LFRS: 80%) to be the preferred options, followed by CCRT+AC, and CCRT alone (**Fig 3**). For prolonging 5-year outcomes, CCRT had the lowest probability of being the preferred option.

**Table 1. Summary of the eligible trials in Bayesian network analysis.**

| Study | Registered number | No. of patients | Stage | Radiotherapy | Induction chemotherapy | Concurrent Chemotherapy | Adjuvant chemotherapy |
|---|---|---|---|---|---|---|---|
| IC+CCRT vs CCRT | | | | | | | |
| Fountzilas 2011 | ACTRN12609000730202 | 141 | AJCC 6th/IIb–IVb | 2DRT/3DRT | cisplatin, epirubicin and paclitaxel | cisplatin 40 mg/m², qw | / |
| Tan 2015 | CDR0000657121 | 172 | UICC 1997/III-IVb | 2DRT/IMRT | gemcitabine, carboplatin and paclitaxel | cisplatin 40 mg/m², qw | / |
| Frikha 2017 | NCT00828386 | 83 | -/T2b, T3, T4 and/or N1-N3, M0 | IMRT/non-IMRT | docetaxel, cisplatin and 5-fluorouracil | cisplatin 40 mg/m², qw | / |
| Hong 2018 | NCT00201396 | 479 | AJCC 5th or UICC 1997/IVa–IVb | 3DRT/IMRT | mitomycin C, epirubicin, cisplatin and 5-fluorouracil | cisplatin 30 mg/m², qw | / |
| Zhang 2019 | NCT01872962 | 480 | AJCC 7th/III–IVb | IMRT | gemcitabine and cisplatin | cisplatin 100 mg/m², q3w | / |
| Sun 2016/Li 2019 | NCT01245959 | 480 | AJCC 7th or UICC 7th/III-IVb, except T3-4N0 | IMRT | docetaxel, cisplatin and 5-fluorouracil | cisplatin 100 mg/m², q3w | / |
| Cao 2017/Yang 2019 | NCT00705627 | 476 | AJCC 6th or UICC 6th/III-IVb, except T3N0-1 | 2DRT/IMRT | cisplatin and 5-fluorouracil | cisplatin 80 mg/m², q3w | / |
| CCRT vs CCRT+AC | | | | | | | |
| Chen 2012/Chen 2017 | NCT00677118 | 508 | AJCC 6th/III-IVb, except T3-4N0 | 2DRT/3DRT/IMRT | / | cisplatin 40 mg/m², qw | cisplatin and 5-fluorouracil |
| IC+CCRT vs CCRT+AC | | | | | | | |
| Lee 2014 | NCT00379262 | 321 | AJCC 6th/III-IVb | 2DRT/3DRT/IMRT | cisplatin and 5-fluorouracil | cisplatin 100 mg/m², q3w | cisplatin and 5-fluorouracil |

Abbreviation: IC, induction chemotherapy; CCRT, concurrent chemotherapy; AC, adjuvant chemotherapy; 2DRT, 2-dimensional radiotherapy; 3DRT, 3-dimensional radiotherapy; IMRT, intensity-modulated radiotherapy; AJCC, American Joint Committee on Cancer; UICC, Union for International Cancer Control.

## Pooled analyses of survival rates

**Fig 4** presented the pooled survival rates in the eligible trials. Among the three treatment strategies, the most efficient therapy for OS was CCRT+AC, followed by IC+CCRT and CCRT. According to this order, the pooled rates of 5-year OS were 83% (95% CI 78%-88%), 80% (95%

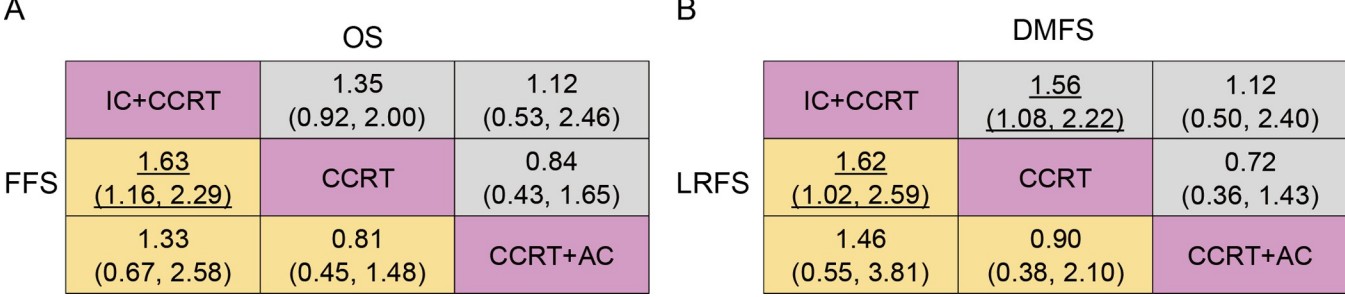

**Fig 2.** Bayesian network analysis results for 5-year overall survival (A up), failure-free survival (A below), distant metastasis-free survival (B up), and locoregional recurrence-free survival (B below). Treatments in the rows were compared with those in the columns. Highlighting numbers, odds ratios (ORs) with significant differences. IC+CCRT, induction chemotherapy followed by concurrent chemoradiotherapy; CCRT+AC, concurrent chemoradiotherapy followed by adjuvant chemotherapy; CCRT, concurrent chemoradiotherapy.

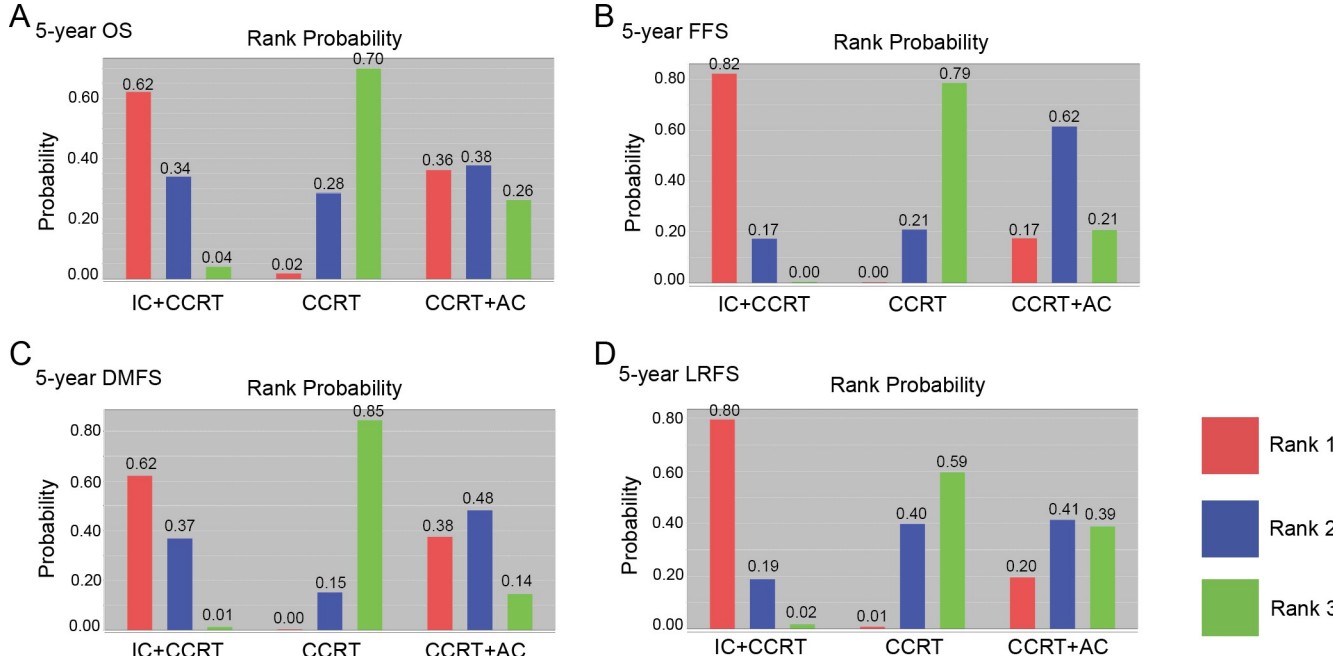

**Fig 3. Rank probabilities of each treatment based on the random-effects model.** (A) 5-year overall survival; (B) 5-year failure-free survival; (C) 5-year distant metastasis-free survival; (D) 5-year locoregional recurrence-free survival.

CI 72%-87%), and 76% (95% CI 71%-81%) (**Fig 4A**). Pooled 5-year FFS rate was 75% (95% CI 70%-80%) in CCRT+AC versus 71% (95% CI 61%-80%) in IC+CCRT versus 63% (95% CI 54%-71%) in CCRT (**Fig 4B**).

The pooled analyzed results of secondary endpoints showed that CCRT+AC (5-year DMFS 85%, 95% CI 80%-89%; 5-year LRFS 91%, 95% CI 87%-94%) had the highest distant metastasis and locoregional recurrence controlled rates compared to IC+CCRT (5-year DMFS 83%, 95% CI 78%-88%; 5-year LRFS 86%, 95% CI 80%-92%) and CCRT alone (5-year DMFS 77%, 95% CI 73%-81%; 5-year LRFS 82%, 95% CI 75%-90%) (**Fig 4C and 4D**).

## Discussion

Our study fully collected the prospective registered and randomized clinical trials and comprehensively analyzed the efficacies of IC+CCRT, CCRT+AC, and CCRT on LANPC. Based on our Bayesian network analysis, IC+CCRT ranked the best for all 5-year survival outcomes. In addition, CCRT+AC showed the highest response rates in the pooled analyses.

### IC+CCRT vs CCRT

Multiple phase III clinical trials have confirmed the critical role of IC+CCRT in treating patients with LANPC. Compared with CCRT alone, IC+CCRT had better HRs in terms of OS, FFS, DMFS, and LRFS [13].

In Zhang's study [21] published in 2019, gemcitabine plus cisplatin as IC combined with CCRT increased the 3-year OS rate to 94.6% versus 90.3% in CCRT group. IC could be well tolerated as 96.7% of the patients in IC+CCRT group completed three cycles of IC. In the newest NCCN guideline [15], gemcitabine and cisplatin regimen was recommended as a category 1 treatment for LANPC patients given IC+CCRT. In addition to this regimen, taxane (including docetaxel, paclitaxel, and nab-paclitaxel) plus cisplatin strategy is used in our hospital. A

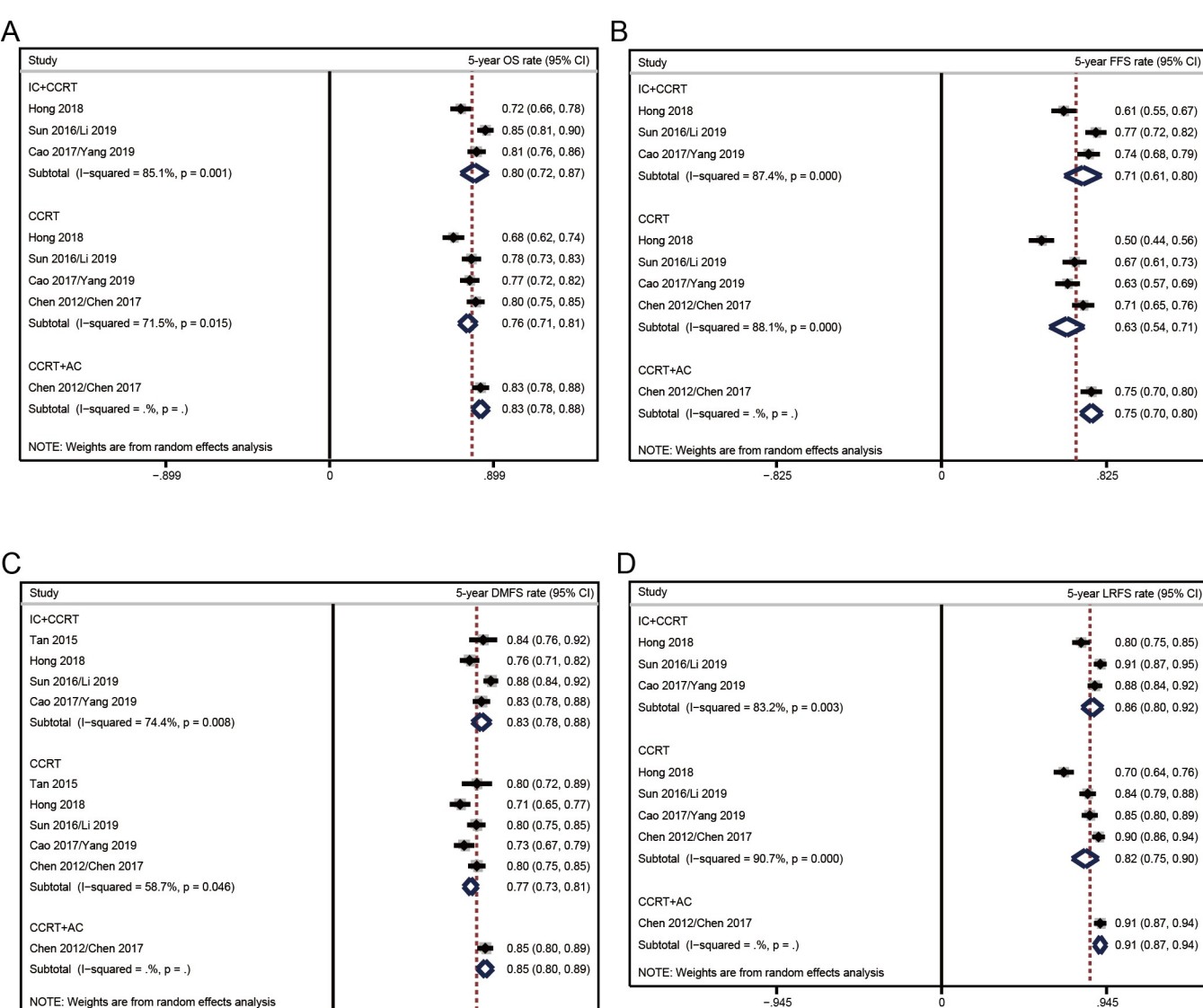

**Fig 4. Pooled results of single-arm data for each therapeutic strategy.** (A) 5-year overall survival; (B) 5-year failure-free survival; (C) 5-year distant metastasis-free survival; (D) 5-year locoregional recurrence-free survival. A dotted line indicated the survival rate analyzed from all the data included in that figure.

propensity-score matching analysis indicated that there were no significant differences in clinical outcomes and safety profiles between docetaxel plus cisplatin and gemcitabine plus cisplatin [26]. Therefore, we suggest taxane or gemcitabine plus cisplatin as a front IC option for LANPC patients.

In addition, Lv et al. conducted a randomized phase 3 clinical trial and compared lobaplatin plus 5-fluorouracil with cisplatin plus 5-fluorouracil in LANPC patients. According to the report, no significant differences were observed between cisplatin-based IC and lobaplatin-based IC in terms of OS (hazard ratio [HR] 0.90, 95% CI 0.55–1.45; $p = 0.65$) and progression-free survival (HR 0.98, 95% CI 0.69–1.39; $p = 0.92$) [27]. Thus, lobaplatin-based strategies might be another promising induction chemotherapies for patients with LANPC.

## CCRT+AC vs CCRT

CCRT+AC failed to show the superiority compared to IC+CCRT or CCRT alone in the Bayesian analysis. However, the survival rates in patients received CCRT+AC were numerically highest. Since most data of CCRT+AC were contributed by Chen's studies [9,10], that is the reason why CCRT+AC did not have higher ORs than the other two therapies. Although there were no significant differences between the groups in Chen's clinical trial, the survival rates in CCRT+AC group were numerically higher than those in CCRT group. Kim and colleagues retrospectively detected the benefit of addition AC to CCRT and found that CCRT+AC showed higher 3-year OS (86% vs 80%, p = 0.894) and FFS (75% vs 66%, p = 0.018) rates against CCRT [28]. However, another retrospective study determined that patients with LANPC might not receive significant survival benefits from adding AC to CCRT (OS HR 0.77, 95% CI 0.37–1.57; FFS HR 1.26, 95% CI 0.69–2.28) [29]. Combined with our Bayesian network analysis, to date, there is still a lack of evidence to confirm the superiority of CCRT+AC compared to CCRT.

In 2021, Hui et al. detected the prediction function of plasma Epstein barr virus (EBV) deoxyribonucleic acid (DNA) and found that the patients with detectable post-radiotherapy plasma EBV DNA who experienced subsequent plasma EBV DNA clearance might be the potential target population of AC [30]. In further explorations, the ongoing phase II and III study, NRG-HN001, attempt to risk-stratify AC using post-radiotherapy plasma biomarker EBV DNA levels. The future results of NRG-HN001 might help us determine the suitable population and whether omitting AC will result in non-inferior survivals compared to patients treated with CCRT+AC.

## IC+CCRT vs CCRT+AC

Large-scale phase III clinical trials that directly compare IC+CCRT with CCRT+AC are limited. NPC-0501 was the only prospective randomized study we identified [25]. Compared with cisplatin plus 5-fluorouracil AC arm (75%), cisplatin plus 5-fluorouracil IC arm had a similar 3-year FFS rate (79%), while cisplatin plus capecitabine arm showed a significantly higher 3-year FFS rate (81%). Therefore, different chemotherapeutic regimens might have impacts on responses. Moreover, NPC-0501 indicated that accelerated fractionation radiotherapy did not achieve more benefits but incurred higher toxicities against conventional fractionation radiotherapy. According to the retrospective results reported by Setakornnukul and colleagues, IC+CCRT was not superior to CCRT+AC (adjusted OS 3-year: 84% vs 81%; 5-year: 74% vs 70%) [31].

Based on the results of published studies and our analysis, IC+CCRT and CCRT+AC display comparable effects on LANPC patients. However, we would recommend IC+CCRT based on the improved probability of being the preferred regimen in all 5-year Bayesian models.

## Limitations

(1) all enrolled clinical trials were open-label studies, which might increase the bias of evaluation; (2) only two trials provided the data for AC, leading to weak evidence for supporting the accurate efficacy of CCRT+AC on LANPC patients; (3) weekly and triweekly cisplatin regimens were conducted during CCRT, however, there is still a debate on the response rates of between the two treatment modalities (week NPC); (4) Analyses of individual patient data are lacking.

## Conclusions

Among the three standard therapeutic strategies, IC+CCRT can be recommended as the preferred choice for previously untreated LANPC patients. Even if current studies suggest CCRT

+AC is not inferior to IC+CCRT, evidence comparing CCRT+AC to the other two strategies remains limited.

## Supporting information

**S1 Checklist. PRISMA checklist.**
(DOC)

**S1 Fig. Network of the comparisons for the Bayesian network meta-analysis.** Network for 5-year overall survival (A), failure-free survival (B), distant metastasis-free survival (C), and locoregional recurrence-free survival (D). each circular node represents a type of treatment. The dot size is proportional to the total number of patients who received a regimen. Each line represents a head-to-head comparison and the line width is proportional to the number of clinical trials comparing the connected treatment strategies.
(DOCX)

**S2 Fig. Risk of bias assessment in the analysis.**
(DOCX)

**S1 File.**
(DOCX)

## Acknowledgments

We thank the members in the SNOWELL STUDIO for helping to improve the manuscript.

## Author Contributions

**Conceptualization:** Zhan-Jie Zhang, Bo-Ya Xiao, Bi-Cheng Wang.

**Data curation:** Bi-Cheng Wang.

**Funding acquisition:** Bi-Cheng Wang.

**Investigation:** Bi-Cheng Wang.

**Methodology:** Bo-Ya Xiao, Guo-He Lin, Bi-Cheng Wang.

**Resources:** Guo-He Lin.

**Software:** Bo-Ya Xiao, Bi-Cheng Wang.

**Supervision:** Quentin Liu.

**Writing – original draft:** Zhan-Jie Zhang, Liang-Liang Shi, Xiao-Hua Hong, Bo-Ya Xiao, Guo-He Lin, Quentin Liu, Bi-Cheng Wang.

**Writing – review & editing:** Zhan-Jie Zhang, Liang-Liang Shi, Xiao-Hua Hong, Bo-Ya Xiao, Guo-He Lin, Quentin Liu, Bi-Cheng Wang.

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
