## [Decision Letter · Decision Letter 0]

23 Feb 2022

PONE-D-21-23917

A Bayesian network analysis of the primary definitive therapies for locoregionally advanced nasopharyngeal carcinoma: IC+CCRT, CCRT+AC, and CCRT alone

PLOS ONE

Dear Dr. Wang,

Thank you for submitting your manuscript to PLOS ONE. After careful consideration, we feel that it has merit but does not fully meet PLOS ONE’s publication criteria as it currently stands. Therefore, we invite you to submit a revised version of the manuscript that addresses the points raised during the review process.

We look forward to receiving your revised manuscript.

Kind regards,

Sethu Thakachy Subha, M.S

Academic Editor

PLOS ONE

Journal Requirements:

2. Please ensure that you have addressed all items recommended in the PRISMA checklist including identifying the study as a meta-analysis or systematic review in the title.

3. Please provide the full electronic search strategy for at least one database, including any limits used, such that it could be repeated

5. Thank you for stating the following financial disclosure: "No"

6. Please include your tables as part of your main manuscript and remove the individual files. Please note that supplementary tables (should remain/ be uploaded) as separate "supporting information" files"

Reviewers' comments:

Reviewer's Responses to Questions

**Comments to the Author**

1. Is the manuscript technically sound, and do the data support the conclusions?

Reviewer #1: Yes

Reviewer #2: Yes

2. Has the statistical analysis been performed appropriately and rigorously? 

Reviewer #1: Yes

Reviewer #2: Yes

3. Have the authors made all data underlying the findings in their manuscript fully available?

Reviewer #1: Yes

Reviewer #2: Yes

4. Is the manuscript presented in an intelligible fashion and written in standard English?

Reviewer #1: Yes

Reviewer #2: Yes

5. Review Comments to the Author

Reviewer #1: Dear PLOS ONE,

It's an honor to review an original manuscript collaborating with your system.

This manuscript was a hard work to provide a great contribution for practice changeable update and hope to apply updates in different guidelines as a better management in locoregional advanced head and neck cancer . A newly diagnosed unresectable locoregional advanced disease or unfit for surgery applied with concurrent systemic therapy followed by radiotherapy is preferred previously. This meta analysis describes and includes all evidence based different applied therapies with comparable selection criteria that established a clear outcome with recognized medical statistical methods. After this practice changeable updated work makes it easy to choose therapy and probably applicable to best care of these patients' kind. 

Thank you editorial board.

Kind regards;

Prem Raj Shrestha

+977984641180

Clinical Hematology and stem cell transplant unit, Department of Medicine

Civil service hospital

Minbhawan, New BaneswarKathmandu Nepal.

Reviewer #2: Zhang et al. present an extensive and rigorous study investigating the Bayesian network analysis involved in the primary definitive therapies for locoregionally advanced nasopharyngeal carcinoma. The question of induction chemotherapy versus adjuvant chemotherapy along with concurrent chemoradiotherapy is an important question to optimize the care of these patients with curable disease. While the experiments are well thought out and expertly executed, some areas of the manuscript require clarification before publication.

Title page

1. The fourth affiliation address “3 State Key Laboratory of Oncology in South China……” should be “4 State Key Laboratory of Oncology in South China……”.

Induction section

2. Paragraph 3: ‘However, it was not until 2018 that the level of evidence for IC+CCRT in the NCCN guideline was adjusted from category 3 to category 2A’ should be replaced by “However, until 2018, the level of evidence for IC+CCRT in the NCCN guideline was adjusted from category 3 to category 2A”.

3. Paragraph 4: “applying” should be replaced by “application”.

Methods section

4. Paragraph 3: “satisfied” should be replaced by “satisfy”.

5. Paragraph 4: “The primary endpoints were 5-year overall survival (OS) and failure-free survival (FFS) rates” could be replaced by “The primary endpoints were the rates of 5-year overall survival (OS) and failure-free survival (FFS)”.

Results section

6. Paragraph 2: “receive” should be replaced by “received”.

Discussion section

7. The study published by Lv et al. has compared lobaplatin plus 5-fluorouracil with cisplatin plus 5-fluorouracil in locoregionally advanced NPC patients. The authors could add the results to the “IC+CCRT vs CCRT” part. (Lancet Oncol. 2021 May;22(5):716-726.)

8. In terms of adjuvant chemotherapy, another study reported by Hui et al. has detected the function of plasma EBV DNA and explored the potential target population. The authors may consider discussing the detailed results in the “CCRT+AC vs CCRT” part. (Clin Cancer Res. 2021 May 15;27(10):2827-2836.)

6. PLOS authors have the option to publish the peer review history of their article (what does this mean?). If published, this will include your full peer review and any attached files.

Reviewer #1: No

Reviewer #2: **Yes: **Yang Chen

---

## [Author Response · Author response to Decision Letter 0]

2 Mar 2022

Response to the reviewers’ comments

(PONE-D-21-23917)

To Reviewer #1

We truly appreciate the reviewer for considering that our manuscript provides a great contribution for the clinical practice in treating locoregionally advanced nasopharyngeal carcinoma patients.

To Reviewer #2

We truly appreciate the reviewer for finding our work “extensive and rigorous”. We thank the reviewer’s encouraging comments and very constructive recommendations. We thus addressed the points accordingly.

Title page

1. The fourth affiliation address “3 State Key Laboratory of Oncology in South China……” should be “4 State Key Laboratory of Oncology in South China……”.

Reply: 

This is a very helpful comment. We are very sorry for the typing mistake. The number of the affiliation address has been corrected. The details are as followings:

Revised Title page (Page 1, line 13 in the revised manuscript):

1 Cancer Center, Union Hospital, Tongji Medical College, Huazhong University of Science and Technology, Wuhan 430022, China

2 Eastern Hepatobiliary Surgery Hospital, Second Military Medical University, Shanghai 200438, China

3 Department of Oncology, the Second Affiliated Hospital of Anhui Medical University, Hefei 230601, China

4 State Key Laboratory of Oncology in South China, Collaborative Innovation Center for Cancer Medicine, Cancer Center, Sun Yat-sen University, Guangzhou 510060, China 

Induction section

2. Paragraph 3: ‘However, it was not until 2018 that the level of evidence for IC+CCRT in the NCCN guideline was adjusted from category 3 to category 2A’ should be replaced by “However, until 2018, the level of evidence for IC+CCRT in the NCCN guideline was adjusted from category 3 to category 2A”.

Reply: 

We thank the reviewer for the very constructive recommendation. The sentence has been adjusted according to the reviewer’s suggestion. The details are as followings: 

Revised Introduction (Page 4, line 16-17 in the revised manuscript):

From 2015, more prospective and large randomized studies have illustrated the benefit of induction chemotherapy followed by CCRT (IC+CCRT) versus CCRT alone in the treatment of LANPC [11]. However, until 2018, the level of evidence for IC+CCRT in the NCCN guideline was adjusted from category 3 to category 2A [12]. Our recently published study also demonstrated that, compared with CCRT, IC+CCRT significantly reduced the risks of death in patients with LANPC (3-year OS hazard ratio [HR]: 0.70, 95% confidence interval [CI] 0.55-0.89; 5-year OS HR: 0.77, 95% CI 0.62–0.94) [13]. To date, when LANPC patients receive IC, gemcitabine/cisplatin or docetaxel/cisplatin/5-fluorouracil could be recommended as category 1 regimens [14, 15]

3. Paragraph 4: “applying” should be replaced by “application”.

Reply: 

We thank the reviewer for this comment. We have replaced “applying” by “application” in the revised manuscript. The details are as followings: 

Revised Introduction (Page 4, line 26 in the revised manuscript):

According to the 2020 NCCN guideline, IC+CCRT, CCRT+AC, and CCRT alone are all the primary definitive therapies for LANPC. Nevertheless, the cited studies recommending the application of CCRT+AC have not been updated. Since the absence of a randomized trial directly comparing IC+CCRT to CCRT+AC and to explore the optimal therapeutic strategy for LANPC, we conducted this Bayesian network analysis to comprehensively compare the efficacies of IC+CCRT, CCRT+AC, and CCRT.

Methods section

4. Paragraph 3: “satisfied” should be replaced by “satisfy”.

Reply: 

This is a helpful comment. We have replaced “applying” by “application” in the revised manuscript. The details are as followings: 

Revised Methods (Page 5, line 19 in the revised manuscript):

Selection criteria

Eligible trials were requested to satisfy the following “PICO” inclusion criteria: (P) patients were newly diagnosed with LANPC; (I+C) LANPC patients randomly received at least two of the three treatment strategies, including IC+CCRT, CCRT+AC, and CCRT; (O) full-text articles and data of survival rates were available. Additional criteria comprised: (1) CCRT was cisplatin-based conventional concomitant chemoradiotherapy; (2) trials should be officially registered prospective phase II-IV clinical studies; (3) target therapy was prohibited during the whole care process; (4) published language was English. Any discrepancies were resolved by discussion.

5. Paragraph 4: “The primary endpoints were 5-year overall survival (OS) and failure-free survival (FFS) rates” could be replaced by “The primary endpoints were the rates of 5-year overall survival (OS) and failure-free survival (FFS)”.

Reply: 

We thank the reviewer for the recommendation. The sentence has been adjusted according to the reviewer’s suggestion. The details are as followings: 

Revised Introduction (Page 5, line 28-29 in the revised manuscript):

Outcomes

The primary endpoints were the rates of 5-year overall survival (OS) and failure-free survival (FFS). The secondary endpoints were 5-year rates of distant metastasis-free survival (DMFS) and locoregional recurrence-free survival (LRFS).

Results section

6. Paragraph 2: “receive” should be replaced by “received”.

Reply: 

Thank you for this helpful comment. We have replaced “receive” by “received” in the revised manuscript. The details are as followings: 

Revised Results (Page 7, line 9 in the revised manuscript):

The basic characteristics of all eligible trials were listed in Table 1. A total of 3140 randomly assigned LANPC patients were involved: 1321 received IC+CCRT, 411 received CCRT+AC, and 1408 received CCRT. 2/3-dimensional radiotherapy (2/3DRT) and intensity-modulated radiotherapy (IMRT) had been applied in these trials. The regimens of induction chemotherapy comprised cisplatin+epirubicin+paclitaxel, gemcitabine+carboplatin+paclitaxel, docetaxel+cisplatin+5-fluorouracil, mitomycin C+epirubicin+cisplatin+5-fluorouracil, gemcitabine+cisplatin, and cisplatin+5-fluorouracil. The adjuvant chemotherapy in both selected trials was cisplatin+5-fluorouracil. In NPC-0501 trial, we extracted data of two arms (patients received cisplatin plus 5-fluorouracil IC or AC) in order to reduce the risk of bias [25]. In addition, the concurrent chemotherapies included weekly and triweekly cisplatin strategies.

Discussion section

7. The study published by Lv et al. has compared lobaplatin plus 5-fluorouracil with cisplatin plus 5-fluorouracil in locoregionally advanced NPC patients. The authors could add the results to the “IC+CCRT vs CCRT” part. (Lancet Oncol. 2021 May;22(5):716-726.)

Reply:

We thank the reviewer for this constructive suggestion. The results reported by Lv et al. has been discussed in the revised manuscript. The details are as followings:

Revised Discussion (Page 9, line 22-28 in the revised manuscript):

IC+CCRT vs CCRT

Multiple phase III clinical trials have confirmed the critical role of IC+CCRT in treating patients with LANPC. Compared with CCRT alone, IC+CCRT had better HRs in terms of OS, FFS, DMFS, and LRFS [13]. 

In Zhang’s study [21] published in 2019, gemcitabine plus cisplatin as IC combined with CCRT increased the 3-year OS rate to 94.6% versus 90.3% in CCRT group. IC could be well tolerated as 96.7% of the patients in IC+CCRT group completed three cycles of IC. In the newest NCCN guideline [15], gemcitabine and cisplatin regimen was recommended as a category 1 treatment for LANPC patients given IC+CCRT. In addition to this regimen, taxane (including docetaxel, paclitaxel, and nab-paclitaxel) plus cisplatin strategy is used in our hospital. A propensity-score matching analysis indicated that there were no significant differences in clinical outcomes and safety profiles between docetaxel plus cisplatin and gemcitabine plus cisplatin [26]. Therefore, we suggest taxane or gemcitabine plus cisplatin as a front IC option for LANPC patients.

In addition, Lv et al. conducted a randomized phase 3 clinical trial and compared lobaplatin plus 5-fluorouracil with cisplatin plus 5-fluorouracil in LANPC patients. According to the report, no significant differences were observed between cisplatin-based IC and lobaplatin-based IC in terms of OS (hazard ratio [HR] 0.90, 95% CI 0.55-1.45; p = 0.65) and progression-free survival (HR 0.98, 95% CI 0.69-1.39; p = 0.92) [27]. Thus, lobaplatin-based strategies might be another promising induction chemotherapies for patients with LANPC.

27. Lv X, Cao X, Xia WX, Liu KY, Qiang MY, Guo L, et al. Induction chemotherapy with lobaplatin and fluorouracil versus cisplatin and fluorouracil followed by chemoradiotherapy in patients with stage III-IVB nasopharyngeal carcinoma: an open-label, non-inferiority, randomised, controlled, phase 3 trial. Lancet Oncol. 2021;22(5):716-26. Epub 2021/04/16. doi: 10.1016/S1470-2045(21)00075-9. PubMed PMID: 33857411.

8. In terms of adjuvant chemotherapy, another study reported by Hui et al. has detected the function of plasma EBV DNA and explored the potential target population. The authors may consider discussing the detailed results in the “CCRT+AC vs CCRT” part. (Clin Cancer Res. 2021 May 15;27(10):2827-2836.)

Reply:

We are thankful for this comment. The results reported by Hui et al. has been added in the revised manuscript. The details are as followings:

Revised Discussion (Page 10, line 14-21 in the revised manuscript):

CCRT+AC vs CCRT

CCRT+AC failed to show the superiority compared to IC+CCRT or CCRT alone in the Bayesian analysis. However, the survival rates in patients received CCRT+AC were numerically highest. Since most data of CCRT+AC were contributed by Chen’s studies [9, 10], that is the reason why CCRT+AC did not have higher ORs than the other two therapies. Although there were no significant differences between the groups in Chen’s clinical trial, the survival rates in CCRT+AC group were numerically higher than those in CCRT group. Kim and colleagues retrospectively detected the benefit of addition AC to CCRT and found that CCRT+AC showed higher 3-year OS (86% vs 80%, p = 0.894) and FFS (75% vs 66%, p = 0.018) rates against CCRT [28]. However, another retrospective study determined that patients with LANPC might not receive significant survival benefits from adding AC to CCRT (OS HR 0.77, 95% CI 0.37-1.57; FFS HR 1.26, 95% CI 0.69-2.28) [29]. Combined with our Bayesian network analysis, to date, there is still a lack of evidence to confirm the superiority of CCRT+AC compared to CCRT.

In 2021, Hui et al. detected the prediction function of plasma Epstein barr virus (EBV) deoxyribonucleic acid (DNA) and found that the patients with detectable post-radiotherapy plasma EBV DNA who experienced subsequent plasma EBV DNA clearance might be the potential target population of AC [30]. In further explorations, the ongoing phase II and III study, NRG-HN001, attempt to risk-stratify AC using post-radiotherapy plasma biomarker EBV DNA levels. The future results of NRG-HN001 might help us determine the suitable population and whether omitting AC will result in non-inferior survivals compared to patients treated with CCRT+AC.

30. Hui EP, Ma BBY, Lam WKJ, Chan KCA, Mo F, Ai QH, et al. Dynamic Changes of Post-Radiotherapy Plasma Epstein-Barr Virus DNA in a Randomized Trial of Adjuvant Chemotherapy Versus Observation in Nasopharyngeal Cancer. Clin Cancer Res. 2021;27(10):2827-36. Epub 2021/03/12. doi: 10.1158/1078-0432.CCR-20-3519. PubMed PMID: 33692028.

To Editor

1. The title has been adjusted to meet the publication criteria. The details are as followings:

A Bayesian network meta-analysis of the primary definitive therapies for locoregionally advanced nasopharyngeal carcinoma: IC+CCRT, CCRT+AC, and CCRT alone

Zhan-Jie Zhang1, Liang-Liang Shi1, Xiao-Hua Hong1, Bo-Ya Xiao2, Guo-He Lin3, Quentin Liu4, Bi-Cheng Wang1,ξ

1 Cancer Center, Union Hospital, Tongji Medical College, Huazhong University of Science and Technology, Wuhan 430022, China

2 Eastern Hepatobiliary Surgery Hospital, Second Military Medical University, Shanghai 200438, China

3 Department of Oncology, the Second Affiliated Hospital of Anhui Medical University, Hefei 230601, China

4 State Key Laboratory of Oncology in South China, Collaborative Innovation Center for Cancer Medicine, Cancer Center, Sun Yat-sen University, Guangzhou 510060, China

---

## [Editor Report · Decision Letter 1]

4 Mar 2022

A Bayesian network meta-analysis of the primary definitive therapies for locoregionally advanced nasopharyngeal carcinoma: IC+CCRT, CCRT+AC, and CCRT alone

PONE-D-21-23917R1

Dear Dr. Wang,

We’re pleased to inform you that your manuscript has been judged scientifically suitable for publication and will be formally accepted for publication once it meets all outstanding technical requirements.

Kind regards,

Sethu Thakachy Subha, M.S

Academic Editor

PLOS ONE
---

## [Editor Report · Acceptance letter]

10 Mar 2022

PONE-D-21-23917R1 

A Bayesian network meta-analysis of the primary definitive therapies for locoregionally advanced nasopharyngeal carcinoma: IC+CCRT, CCRT+AC, and CCRT alone 

Dear Dr. Wang:

I'm pleased to inform you that your manuscript has been deemed suitable for publication in PLOS ONE. Congratulations! Your manuscript is now with our production department. 

Kind regards, 

on behalf of

Dr. Sethu Thakachy Subha 

Academic Editor

PLOS ONE